# LEARNING REPRESENTATIONS FOR FASTER SIMILARITY SEARCH

## ABSTRACT

In high dimensions, the performance of nearest neighbor algorithms depends crucially on structure in the data. While traditional nearest neighbor datasets consisted mostly of hand-crafted feature vectors, an increasing number of datasets comes from representations learned with neural networks. We study the interaction between nearest neighbor algorithms and neural networks in more detail. We find that the network architecture can significantly influence the efficacy of nearest neighbor algorithms even when the classification accuracy is unchanged. Based on our experiments, we propose a number of training modifications that lead to significantly better datasets for nearest neighbor algorithms. Our modifications lead to learned representations that can accelerate nearest neighbor queries by $5\times$.

## 1 INTRODUCTION

Learning representations has become a major field of study over the past decade, with many succesful applications in computer vision, speech recognition, and natural language processing. The primary focus in these directions has been *accuracy*, usually with a focus on classification tasks. Here, the main goal is to learn a representation that enables a standard classifier (e.g., a linear model such as softmax regression) to correctly label the transformed data. However, as learned representations have achieved high accuracy in a wide range of applications, additional goals are becoming increasingly important. One such desideratum is *computational efficiency*: how quickly can we process the learned representations? This is question is particularly relevant in the context of large databases, where the goal is to store many millions or even billions of images, texts, and videos. Common instantiations of such settings include web search, recommender systems, near-duplicate detection (e.g., for copyrighted content), and large-scale face recognition.

In this paper, we study the problem of learning representations through the lenss of *similarity search*. Similarity search, also known as Nearest Neighbor Search (NNS), is a fundamental algorithmic problem with a wide range of applications in machine learning and broader data science . The most common example is similarity search in large corpora such as the aforementioned image databases, segments of speech, or document collections. More recently, NNS has also appeared as a sub-routine in other algorithms such as optimization methods (Dhillon et al., 2011), cryptography (Laarhoven, 2015), and large-scale classification (Vijayanarasimhan et al., 2015). A key challenge in the design of efficient NNS methods is the interaction between the data and the algorithms: How can we exploit structure in the data to enable fast and accurate search? Research on NNS has established a large set of techniques such as kd-trees, locality-sensitive hashing, and quantization methods that utilize various forms of structure in the data.

Traditionally, NNS is used on hand-crafted feature vectors: image similarity search is often performed on SIFT vectors, speakers are identified via their i-vector, and document similarity is computed via tf-idf representations. However, recent progress in deep learning is replacing many of these hand-crafted feature vectors with learned representations. There is often a clear reason: learned representations often lead to higher accuracy and semantically more meaningful NNS results. However, this raises an important question: Do existing NNS algorithms perform well on these new classes of feature vectors? Moreover, learning the representations in NNS also offers an interesting new design space: Instead of adapting the algorithm to the dataset, can we learn a representation that is particularly well suited for fast NNS?

Our main contribution is to study this interaction between deep representations and NNS algorithms in more detail. First, we point out that angular gaps are a crucial property of learned representations when it comes to the efficiency of NNS methods. We then explore the design space of neural networks and find that architectural changes such as normalization layers can have significant impact on the angular gaps, even when the classification accuracy is not affected. Based on our experiments, we propose changes to the network architecture and training process that make the resulting representations more amenable to NNS algorithms. Our proposals are intuitive and simple to implement, yet enable speed-ups of $5\times$ or more for nearest neighbor search. Moreover, our changes do not negatively affect the accuracy of the network and sometimes even improve training.

**Paper outline.**  In Section 2, we first explain a property of vectorial representations that is crucial for the performance of NNS methods. This property will guide our evaluation of various neural network design consideration in Section 3. We present our experimental results in Section 4.

## 2  SMALLER ANGLES, FASTER SIMILARITY SEARCH

We first take a closer look at what properties of representations enable fast nearest neighbor search (NNS). We comprehensively address this question both from a theoretical and an empirical perspective. The resulting insights will then guide our approach to learning representations that are good for NNS.

**Preliminaries.**  Before we begin with our investigations, we briefly define the NNS problem formally. The goal in NNS is to first preprocess a dataset $D$ of points from a metric $(X, \mathrm{dist})$ so that we can later quickly answer queris of the following form: "Given a query point $\mathbf{q} \in X$, which point $\mathbf{p} \in D$ minimizes the distance $\mathrm{dist}(\mathbf{q}, \mathbf{p})$?". A common concrete instance of the NNS problem is similarity search under cosine similarity. Here, the set of points $D$ contains unit vectors in $\mathbb{R}^d$ and the distance measure dist is the angular distance $\angle(\mathbf{q}, \mathbf{p}) \stackrel{eq}{=} \cos^{-1} \frac{\mathbf{q}^\top \mathbf{p}}{\|\mathbf{q}\|_2 \|\mathbf{p}\|_2}$

### 2.1  THEORETICAL ANALYSIS

Different NNS algorithms rely on different properties on the data in order to enable faster similarity search. In this paper, we focus on locality-sensitive hashing (LSH), a well-establish NNS method, where it is possible to analyze the impact of *distance gaps* on the query time quantitatively (Indyk & Motwani, 1998; Har-Peled et al., 2012). Concretely, we study the cosine similarity NNS problem mentioned above and analyze the running time of the popular hyperplane LSH (Charikar, 2002).

Suppose that a query $q$ makes an angle of $\alpha$ with the closest vector. Under reasonable assumptions on the balance of the data, we show the following statement in Appendix C. If we want to ensure that the probability of finding the closest vector is at least, $1 - \exp(-5) \approx 99.33\%$, the expected number of candidates considered in a nearest neighbor search, for a dataset of size $n$, grows with $\alpha$ as

$$N(\alpha) \;=\; 5n^{-\log_2(1 - \frac{\alpha}{\pi})} \tag{1}$$

We defer our formal theorem with its proof to the supplementary material.

Equation (1) allows us to estimate the number of nearest neighbor candidates (i.e., the time complexity of a query) as we vary the angle between the query vector and the nearest neighbor. A crucial property of this relation is that smaller angles between nearest neighbor and the query point enable significantly faster NNS. We illustrate by substituting concrete numbers. For $n = 10^6$ data points, improving the angle $\alpha$ from $\pi/3$ (60 degrees) to $\pi/4$ (45 degrees) reduces the number of nearest neighbor candidates (i.e., distance computations) from about 16K to 1.5K, whichis roughly a $10\times$ speed-up.. In Figure 1(a), we plot the expression (1) as a function of the angle $\alpha$ for $n = 10^6$.

### 2.2  EMPIRICAL ANALYSIS

While the above theoretical analysis is for the relatively simple hyperplane LSH, the same trend also applies to more sophisticated NNS algorithms. We show that two state-of-the-art NNS methods show empirical behavior that is very similar to Figure 1(a). Concretely, we compare the following two ANN implementations that are commonly used:

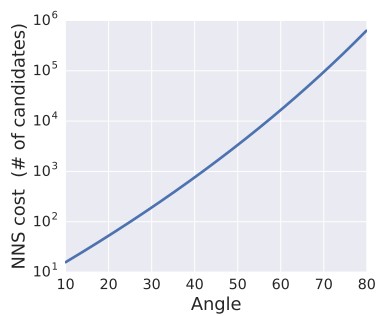
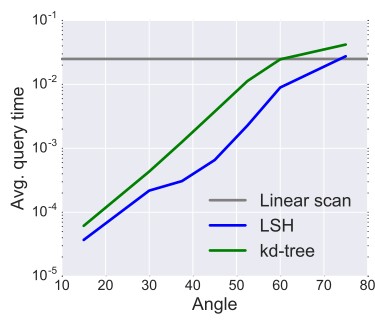

(a) Analytically estimated NNS cost      (b) Empirically estimated NNS cost

Figure 1: Analytically and empirically estimated costs of an approximate nearest neighbor search as a function of the angle between the query and the nearest neighbor. The analytical cost is given as the number of distance computations between the query point and the candidate points returned by the LSH table. The empirical cost is given as the total time per query on a laptop computer from 2015 (CPU: Intel Core i7-4980HQ).

- Annoy, which implements a variant of the classical kd-tree (Bernhardsson, 2013).
- FALCONN, which implements the fastest known LSH family for angular distance (Razenshteyn & Schmidt, 2015).

We generate a dataset of $n = 10^6$ random unit vectors and plant queries at a given angle $\alpha$ from a randomly selected subset of database points. Since the dataset is drawn from the uniform distribution over the hypersphere, it is "well-spread" and matches the assumptions of our analysis above. For each angle $\alpha$, we generate 10,000 queries and report the average query time for finding the correct nearest neighbor with empirical success probability 0.9.

Figure 1(b) shows that the query times of the two ANN implementations largely agree with our theoretical prediction.[1] An important aspect of the plot is the linear scan baseline, i.e., the cost of computing all distances between query and database points (which always finds the correct nearest neighbor). At 90% relative accuracy, the ANN algorithms we evaluated only improve over a linear scan once the angle is around 60 degrees. For larger angles, current ANN algorithms cannot maintain both high accuracy and faster inference speed. Overall, our empirical findings also underline the importance of the angle between query and nearest neighbor for NNS performance.

## 3 LEARNING REPRESENTATIONS FOR SMALLER ANGLES

In the previous section, we have seen that the angle between the query and its nearest neighbor play a crucial role in the performance of fast NNS methods. We now build on this insight to learn representations where this angle is small. As deep neural networks have become widely used for learning representations, we focus specifically on how to modify neural networks and their training process to enable faster NNS.

### 3.1 PRELIMINARIES

We study neural networks for supervised classification into one of $C$ classes. Each example $\mathbf{x}$ comes from a domain $\mathcal{X}$, typically $\mathbb{R}^p$, and our goal is to predict its label $y \in [C]$. Given training examples $(\mathbf{x}_i, y_i) \in \mathcal{X} \times [C]$, we learn model parameters by optimizing a certain loss function on the training data. We then use the resulting model to predict labels.

For our purposes, a neural network consists of three sets of parameters: (i) the network weights $\theta \in \Theta$ defining a mapping $\phi_\theta : \mathcal{X} \to \mathbb{R}^d$, (ii) a set of *class vectors* $\{\mathbf{v}_y\}_{y \in [C]}$ that also lie in $\mathbb{R}^d$, and (iii)

---

[1]There are some mismatches, for instance in the regime of small angles. For FALCONN, this is because a real LSH implementation also needs to compute the hash functions, which is not included in our theoretical analysis above.

a bias vector $\mathbf{b} \in \mathbb{R}^C$. We will often write the class vectors as a matrix $V$. We refer to the vectors $\phi_\theta(\mathbf{x})$ as the *representation vectors*. We employ the commonly used softmax loss, which is defined as:

$$\ell(\mathbf{x}, y; \theta, V, \mathbf{b}) \overset{eq}{=} -\log \frac{\exp(\phi_\theta(\mathbf{x})^\top \mathbf{v}_y + b_y)}{\sum_{j \in [C]} \exp(\phi_\theta(\mathbf{x})^\top \mathbf{v}_j + b_j)} .$$

Noting that $\phi_\theta(\mathbf{x})^\top \mathbf{v}_j + b_j = (\phi_\theta(\mathbf{x}), 1)^\top (\mathbf{v}_j, b_j)$, the bias term is superflous for the purposes of the softmax objective. To simplify the discussion, we will therefore ignore the biases and write:

$$\ell(\mathbf{x}, y; \theta, V) \overset{eq}{=} -\log \frac{\exp(\phi_\theta(\mathbf{x})^\top \mathbf{v}_y)}{\sum_{j \in [C]} \exp(\phi_\theta(\mathbf{x})^\top \mathbf{v}_j)} .$$

**The Softmax Objective.**  To understand the softmax objective better, we write it as

$$\ell(\mathbf{x}, y; \theta, V) = -\phi_\theta(\mathbf{x})^\top \mathbf{v}_y + \log \sum_{j \in [C]} \exp(\phi_\theta(\mathbf{x})^\top \mathbf{v}_j).$$

The training process aims to achieve a small loss, i.e., we want the correct dot product $\phi_\theta(\mathbf{x})^\top \mathbf{v}_y$ to be large and the other dot products $\phi_\theta(\mathbf{x})^\top \mathbf{v}_j$ to be small. We further re-write the inner product in order to highlight the connection to the angle between class vectors and representation vectors:

$$\phi_\theta(\mathbf{x})^\top \mathbf{v}_j = \|\phi_\theta(\mathbf{x})\|_2 \, \|\mathbf{v}_j\|_2 \, \cos \angle(\phi_\theta(\mathbf{x}), \mathbf{v}_j) . \tag{2}$$

Viewed this way, three properties of the loss function become apparent:

1. While the relative norm of the class vectors does influence the predictions of the model, the overall scale does not. Doubling the lengths of all class vectors has no influence on the prediction accuracy, yet it can change the softmax loss.

2. The norm of the representation vector $\phi_\theta(\mathbf{x})$ is irrelevant for the 0-1 prediction loss objective. Yet it can have significant influence on the softmax loss.

3. A small angle between the representation vector $\phi_\theta(\mathbf{x})$ and the class vector $\mathbf{v}_j$ is beneficial for both the prediction loss objective as well as the softmax loss objective. Other things being equal, we prefer a model that maps examples to representation vectors that are well-aligned with the correct class vectors.

Next section, we view these three terms through the lens of angles for NNS problems.

## 3.2  IMPROVING THE REPRESENTATION QUALITY: SMALLER ANGLES

When training a representation with small angles between query and nearest neighbor, we will view the representation before the softmax layer as the query point and the class vectors in the softmax as dataset points. Our main focus is then to improve the *mean correct angle* between the representation vector $\phi_\theta(\mathbf{x})$ and the correct class vector $\mathbf{v}_y$. For an evaluation set $\{(\mathbf{x}_1, y_1), \ldots, (\mathbf{x}_m, y_m)\}$, we define the mean correct angle as

$$\frac{1}{m} \sum_{i=1}^{m} \angle(\phi_\theta(\mathbf{x}_i), \mathbf{v}_{y_i}) .$$

In this section, we describe how our modifications improve the mean correct angle. As we have seen in Section 2, this angle is an algorithm-agnostic quantity that dictates the speed-ups we can expect from various ANN algorithms.

To illustrate the impact of various network design choices, we evaluate the effect of each modification on a medium-sized dataset (in the supplementary material, we also show the effect on a smaller dataset that facilitates easier visualization and aids intuition). In Section 4, we then evaluate our modifications on two larger datasets.

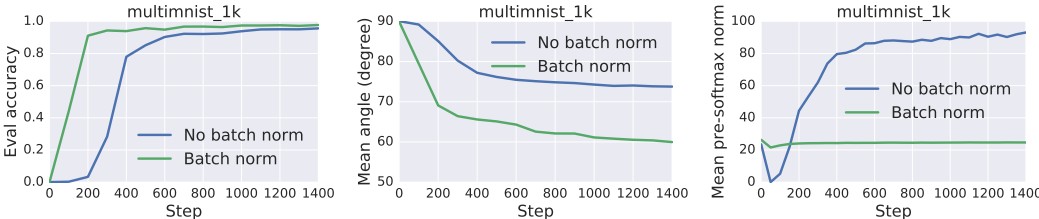

Figure 2: The effect of batch normalization on the evaluation accuracy, mean correct angle, and norm of the representation (pre-softmax) vectors. The plots show two training runs on our multiMNIST dataset with 1,000 classes. The norm of the representation vectors is now nearly constant, and the mean correct angle improves significantly. The normalization also improves training speed.

**multiMNIST.**    This dataset is derived from the popular MNIST dataset (LeCun & Cortes, 2010). We construct large-multiclass variants of MNIST by concatenating multiple digits (see Section 4 for more details). We call these variants *multiMNIST*. The multiMNIST instances allow us to confirm our intuitions on a larger dataset with a higher-dimensional softmax and more classes. Here, we use multiMNIST instances with 1,000 to 10,000 classes.

We investigate the impact of three training modifications.

**Control Representation Vector Norms**    As discussed in Section 3.1, scaling the representation vectors $\phi_\theta(\mathbf{x})$ can change the loss function without changing the prediction accuracy. The dynamics of (stochastic) gradient descent update parameters so as to increase the inner product $\phi_\theta(\mathbf{x})^\top \mathbf{v}_j$, or equivalently, the product

$$\|\phi_\theta(\mathbf{x})\|_2 \cdot \|\mathbf{v}_j\|_2 \cdot \cos \angle(\phi_\theta(\mathbf{x}), \mathbf{v}_j) \ .$$

With a constraint on the term $\|\phi_\theta(\mathbf{x})\|_2$, the training process tends to increase the remaining terms instead, which leads to smaller angles $\angle(\phi_\theta(\mathbf{x}), \mathbf{v}_j)$. We consider several options for controlling the norms of the representation vectors. Layer Normalization (Ba et al., 2016) ensures that the presoftmax vector has mean entry zero and a constant norm. Scaling (which is similar to Weight Normalization (Salimans & Kingma, 2016)) divides each presoftmax vector by its mean before propogating and thus ensures a constant norm (but not necessarily mean zero). A variant of scaling just uses the normalized vector but does not back-prop through the normalization. Batch normalization (Ioffe & Szegedy, 2015) can be used to ensure that each activation has mean 0 and variance 1 at equilibrium, which implies that the mean squared norm of the representation is $d$. Finally, one can simply "regularize" by adding a $\lambda \frac{1}{n} \sum_{i=1}^n \|\phi_\theta(\mathbf{x}_i)\|_2^2$ term to the loss.

While batch norm is often used in networks for other reasons, it is unclear *a priori* which of these options would work best from the viewpoints of the angles, the final accuracy and the convergence speed. We compare these options in more detail in AppendixD.2. Our experiments show that from the point of view of accuracy, convergence speed and angles, batch norm empirically gives better results than the other options.

For lack of space, we only show the comparison with batch norm in the rest of this section. Using batch norm does not hurt the accuracy on the evaluation set, nor the convergence speed. The mean angle improves and the representation vector norms are now significantly smaller than in the unconstrained case.

**Use All Directions**    In a standard ReLU-based neural network, the representation vector $\phi(\mathbf{x})$ is the output of a ReLU. In particular, this means that each coordinate of $\phi(\mathbf{x})$ is non-negative, i.e., the representation vector always lies in the non-negative orthant. While the non-linearity in neural networks is an important source of their representation power, the ReLU at the last layer yields a more difficult dataset for NNS methods. Instead of producing a representation on the entire unit sphere, the representation vector is restricted to lie in the non-negative orthant. Implicitly, this also restricts the locations of the class vectors in the softmax: in order to achieve reasonable angle, the class vectors must be close to the non-negative orthant. To ameliorate this issue, we propose to either

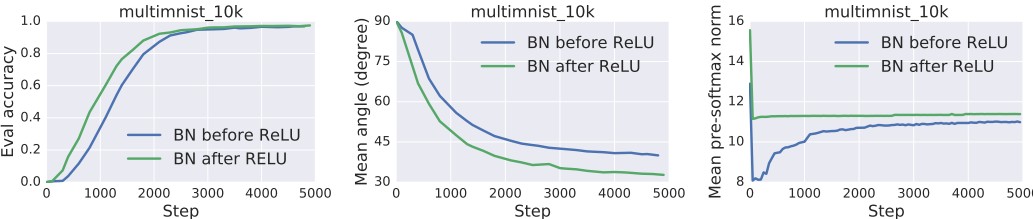

Figure 3: The effect of swapping ReLU and batch normalization before the softmax layer. The plots show two training runs on our multiMNIST dataset with 10,000 classes. The norm of the representation vectors show less variation during training, and the mean correct angle improves. The normalization also improves training speed.

remove the last layer ReLU or to place it *before* the batch norm so that the final representation vector $\phi_\theta(\mathbf{x})$ can have both positive and negative coordinates.

As we can see in Figure 3, this modification improves the mean correct angle on multiMNIST. Moreover, the network achieves a good classification accuracy faster, and the representation vector norms are better controlled.

**Equalize Class Vector Lengths**    NNS is often an easier problem when all dataset points have the same norm as various distance measure such as Euclidean distance, maximum inner product, and cosine similarity are then comparable. Hence it is desirable to ensure that all class vectors in the softmax layer have the same norm. In many trained networks, the ratio between largest and smallest class vector norm is already fairly small, e.g., about $1.5$ in case of the Inception architecture (Szegedy et al., 2015) for Imagenet. The fact that this ratio is not too large suggests that forcing it to be exactly $1$ *while training* may not hurt the performance of the model. This is our third modification: we constrain the norms of the class vectors to all be equal. This constraint can be realized via projected gradient descent: we first initialize the class vectors to have unit norm.[2] Then after every gradient step (or a small number of steps), we re-normalize the class vectors to be norm $1$ again.

This modification has two advantages. First, it guarantees that the NNS problem is exactly an angular NNS problem. Moreover, the modification also helps in achieving smaller angles. Intuitively, we constrain the second term $\|\mathbf{v}_j\|_2$ in Equation (2) to be $1$. We see the effect of this modification on the multiMNIST dataset in Figure 4. The mean correct angle improves and the class vector norms are now significantly smaller than in the unconstrained case. Importantly, the improvement in mean correct angle is *cumulative* with the previous modifications. Overall, we not only obtain an angular NNS problem (instead of MIPS), we also have a better conditioned problem due to the smaller angles.

## 4    EXPERIMENTS

We now evaluate our training modifications on two large datasets. We begin by describing these two datasets that we use in our experiments, together with the corresponding network architecture. In Appendix D, we also report results on a third dataset (CelebA).

**MultiMNIST**    As in Section 3, we use our multiclass variant of the classical MNIST dataset LeCun & Cortes (2010). MNIST contains grayscale images of hand-written digits $\{0, 1, 2, \ldots, 9\}$ with fixed size $28 \times 28$. To create a multiclass dataset, we horizontally concatenate $c$ of these images together to form composed images. We label each composed image with the concatenation of the corresponding digits. Thus, for example $c = 2$ corresponds to class space $\{00, 01, \ldots, 99\}$ and image size $28 \times 56$. We construct a training dataset by concatenating random images from the MNIST train dataset, and proceed similarly for the evaluation dataset. We refer to a multiMNIST dataset with $C$ classes as multiMNIST_$C$. Figure 9 shows an example image from multiMNIST_100K (so $c = 5$).

---

[2]The constant 1 is somewhat arbitrary and other values may be considered.

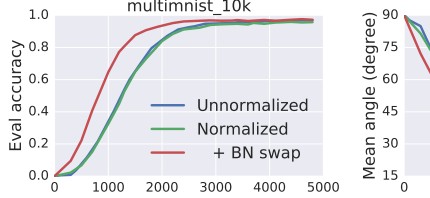 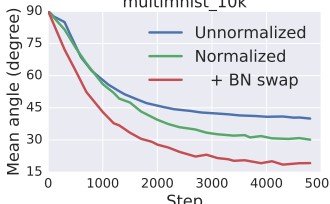 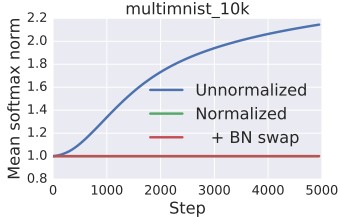

Figure 4: The effect of normalizing the class vectors in the softmax layer. The plots show two training runs on our multiMNIST dataset with 10,000 classes. The average norm of class vectors is now constant and significantly smaller than before. Moreover, the mean correct angle improves while the accuracy is not negatively affected. Importantly, the positive effects are cumulative with the previous modification of swapping the order of ReLU and batch normalization (we are not comparing to a baseline without batch normalization because we could not train a 10,000 class multiMNIST dataset without batch normalization).

Our model architecture is based on the MNIST model architecture in the TensorFlow tutorial Abadi et al. (2015).[3] The network consists of two $5 \times 5$ convolutional layers with 32 and 64 channels, respectively. Both convolutional layers are followed by batch norm, ReLu and MaxPool. We then use a fully connected layer to project down to $d$ dimensions, where $d$ is the softmax dimension typically taken to be between $256$ and $1024$. We train the network with stochastic gradient descent on a single machine with a GPU.

**Sports1M** We perform our second set of experiments on the Sports1M dataset Karpathy et al. (2014).[4] This dataset contains about 1 million videos. As in Vijayanarasimhan et al. (2015), we construct a multiclass problem where the examples are the video frames, and the labels are the video ids. We use the first half of each video as training examples and the second half as test examples. After removing some videos to ensure a common format, this gives us a classification problem with about 850,000 labels.

We convert each frame to its VGG features Simonyan & Zisserman (2014), using a pretrained VGG16 network.[5] In particular, we use the output of the last convolutional layer as features, followed by a fully connected layer of size $1024$ and a softmax layer of dimension $d = 256$. To accelerate training, we employ a sampled softmax loss approximation with sample size 8192. This also allows us to test whether our training modifications are compatible with the sampled softmax approximation (the multiMNIST test cases are trained with a full softmax). We train our models using a distributed TensorFlow setup employing 15 workers with one GPU each and 10 parameter servers.

**Evaluation Objectives** The goal of our experiments is to investigate two main questions: (i) How effective are our modifications for improving the angle between the representation vectors and the class vectors? (ii) Do our modifications hurt the accuracy of the model?

Moreover, we also investigate the "well-distributedness" of the class vectors. In Theorem 1 in Appendix C, we define a specific measure of imbalance that impacts the run time of an LSH-based nearest neighbor scheme. More precisely, we show that as long as second moment of the table loads, denoted by $M(P, h)$ remains close to an "ideal" value $M^*$, smaller angles improve the NNS performance. We want to determine if this measure of imbalance, $M(P, h)/M^*$, remains small.

**Results** The results of our experiments on multiMNIST_100K are shown in Figure 5. The baseline training approach uses the common configuration of applying batch normalization before the nonlinearity and does not normalize the softmax vectors. Compared to this baseline, our training modifications decrease the mean correct angle from about 40 to 15. This corresponds to a $30\times$ faster softmax evaluation at inference time. Moreover, our modifications lead to no loss in accuracy.

---

[3]Available at `https://www.tensorflow.org/get_started/mnist/pros`.
[4]Available at `http://cs.stanford.edu/people/karpathy/deepvideo/`.
[5]Downloaded from `https://www.cs.toronto.edu/~frossard/post/vgg16/`.

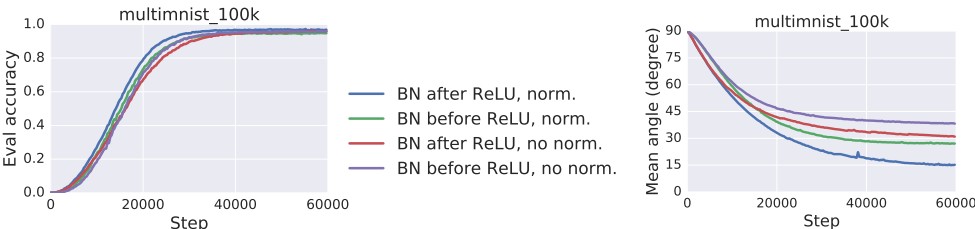

Figure 5: The effect of our training modifications on a multiMNIST_100K dataset. The mean correct angle improves significantly while the evaluation accuracy is not affected negatively.

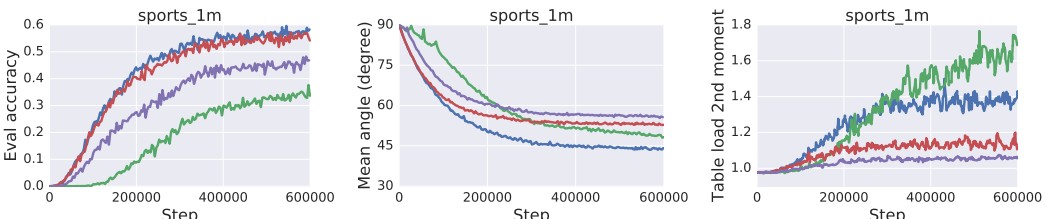

Figure 6: The effect of our training modifications on the Sporst1M dataset (the legend is the same as in Figure 5 above). The mean correct angle is significantly smaller. The evaluation accuracy also improves. Moreover, the points remain evenly spread over the unit sphere, i.e., the "table balance" quantity $M(P, h)/M^*$ remains close to 1, especially for the best-performing combination of our techniques.

Figure 6 shows the results on the Sports1M dataset. In this case, our modifications improve the accuracy after 600K steps. Placing the batch normalization after the non-linearity seems particularly important. The mean correct angle decreases by slightly more than 10 degrees, corresponding to a $10\times$ improvement in inference speed.

The right panel in Figure 6 shows the second moment of the table loads relative to the ideal value, i.e., $M(P, h)/M^*$. While our modifications lead to an increase in this value, the increase is small and stays around $1.4\times$ for the relevant configuration that achieves the best accuracy and mean correct angle.

In summary, our experiments demonstrate that our modifications obtain significantly smaller angles between representation vectors and the correct class vectors. This translates to an order of magnitude improvement in inference speed, and leads to learning better representations. The improvement comes at no decrease in overall accuracy.

## 5 CONCLUSIONS AND FUTURE WORK

We have demonstrated how to learn representations specifically for faster similarity search in large datasets. To this end, we have studied multiple modifications to neural network training and architecture that lead to a smaller angle between the learned representation vectors produced by the network and the class vectors in the softmax layer. This angle is a crucial measure of performance for approximate nearest neighbor algorithms and enables a $5\times$ speed-up in query times. An interesting direction for future research is whether these insights can also lead to faster training of large multiclass networks, which are common in language models or recommender systems.

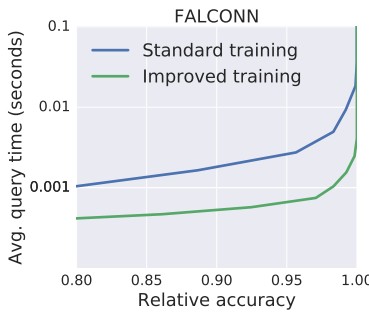 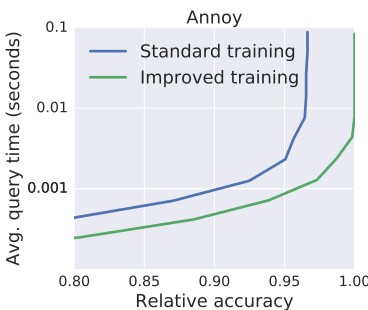

Figure 7: Effect of our training modifications on the query times of nearest neighbor algorithms. We report the *relative* accuracies, i.e., the probability of finding the correct nearest neighbor conditioned on the model being correct. For LSH as implemented in the FALCONN library (left), our training yields a $5\times$ speed-up in the relevant high accuracy regime. The variant of kd-trees implemented in the Annoy library (right) does not reach relative accuracy 1 when the softmax is trained using the standard approach. In contrast, the softmax resulting from our training techniques is more amenable to the kd-tree algorithm. Again, we obtain faster query times for fixed accuracy.

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

## A    RELATED WORK

The paper (Liu et al., 2016) also proposes a method for training with larger angular distance gaps. In contrast to our approach, the authors modify the loss function, not the training process or network architecture. The focus of their paper is also on classification accuracy and not fast similarity search. The authors do not quantify the improvements in angular distance and train on datasets with a relatively small number of classes (100 or less).

**?** also modifies the loss function for learning representations with angular distance gaps. Again, the focus of their paper is on accuracy and not fast similarity search. In particular, the authors do not investigate the effect of their changes on the angular gap on large datasets. The focus of our paper is on fast similarity search and we evaluate various network modifications with end-to-end experiments using state-of-the art NNS methods.

## B    VISUALIZATION OF OUR APPROACH

**2D-GMM.**    To visualize the effect of our modifications, we consider a synthetic example that allows us to illustrate the geometry of the softmax layer. The data is drawn from a small number of well separated Gaussians in two dimensions. The labels correspond to the Gaussians they are sampled from. The network we train has two hidden layers with ten hidden units each, followed by a projection to two dimensions. We use ReLU non-linearities. We train the network using AdaGrad Duchi et al. (2011) to convergence, i.e., (near) zero training loss. The appealing property of this synthetic example is that we can plot the representation vectors $\phi_\theta(\mathbf{x})$ and class vectors $v_i$ directly in two dimensions.[6] The results are in line with those on multimnist and are meant to help visualize the effect of our modifications.

We show in Figure 8 the effect of our modifications on this dataset. In Figures 8(a) and (b). we see that applying batch normalization leads to pre-softmax vectors that have approximately the same norm. Moreover, the angle between the pre-softmax vectors and the correct class vector improves noticeably.

---

[6]A complementary approach would be to train a higher-dimensional softmax and project the vectors to two dimensions for visualization. However, such a projection necessarily distorts the geometry. Hence we have trained a two-dimensional softmax to ensure a faithful illustration.

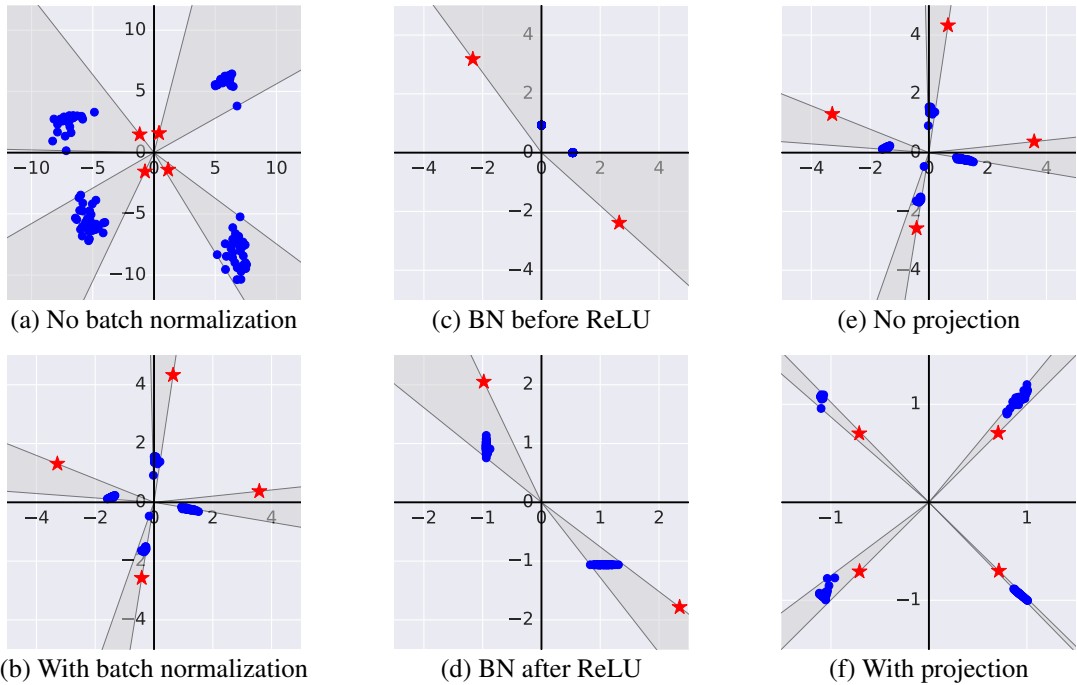

(a) No batch normalization    (c) BN before ReLU    (e) No projection

(b) With batch normalization    (d) BN after ReLU    (f) With projection

Figure 8: Visualization of our training modifications for a two-dimensional softmax trained on a synthetic dataset. Each column shows the effect of one modification. The top row contains the baseline result without the respective training modification and the bottom row shows the result with modification.

The data is drawn from a mixture of Gaussians with equally spread means and variance 0.1 in each direction. Since a ReLU unit before the softmax limits the expressivity of the model, we could only train two classes with the two-dimensional softmax and a final ReLU (middle column). Hence we did not employ a ReLU before the softmax layer in the left and right columns. In all cases, the network trained to accuracy 1.

The blue points in the plots are the representation vectors of the examples and the red ⋆'s are the class vectors in the softmax. The gray sectors indicate the maximum angles between the representation vectors and the corresponding class vectors. The plots show that our modifications lead to significantly smaller angles.

Figures 8(c) and (d) show the effect of using all directions on the 2D-GMM dataset for a mixture of 2 Gaussians. When the ReLU is after the batch norm, the pre-softmax vectors are in the non-negative quadrant. Moving the batch norm after the ReLU leads to smaller angles.

Finally, we visualize the effect of projecting the softmax vectors in Figures 8(e) and (f). This modification leads to even thinner sectors, i.e., even smaller angles between the representation vectors and the correct class vectors.

## C    THEORETICAL ANALYSIS OF NNS

For the case of locality-sensitive hashing (LSH), we can analyze the speed vs. accuracy tradeoff quantitatively Indyk & Motwani (1998); Har-Peled et al. (2012).

Suppose that we use a hyperplane LSH scheme with $k$ hyperplanes per table and $m$ independent hash tables Charikar (2002). For the purpose of our analysis, we consider a variant where we select a candidate set of class vectors as follows: we first find the hash bucket of the query vector in each of the $m$ tables and collect all database points in these hash buckets as a candidate set. Among these candidates, we then compute exact distances in order to find the (approximate) nearest neighbor. The goal is to choose the parameters $m$ and $k$ so that (i) the probability of finding the closest vector in the candidates set is close to 1, and (ii) the number of candidates is small.

In order to analyze this LSH scheme, we need to consider two questions:

- How many hash tables $m$ do we need to query so that we find the nearest neighbor in one of them with good probability?
- How many spurious candidate vectors will we encounter in the $m$ hash buckets we explore?

We now address the first question. For a query $q$, the probability that that a dataset point within an angle of $\alpha$ lies in the same LSH bucket as $q$ is $p_\alpha \stackrel{eq}{=} (1 - \frac{\alpha}{\pi})^k$. To ensure that the failure probability is small, say, $\exp(-5) \approx 0.67\%$, it suffices to take $m$ to be $5/p_\alpha$. It can then be easily verified that we should set $k$ to be as large as possible in order to obtain a small candidate set. However, taking $k$ to be larger than $\log n$ leads to many hash buckets being empty, which increases the cost of generating the set of candidates itself. So we set $k$ to be $\lfloor \log n \rfloor$; see also (Har-Peled et al., 2012, Theorem 3.4).

Next, we bound the total number of candidates in the hash buckets. The expected number of points in a given hash bucket is $n/2^k$ (the expectation is taken over the randomness in the hash function). However, this quantity does not necessarily bound the expected number of candidates during a query. The event that $q$ lands in a certain hash bucket may be correlated with a large number of database points occupying the same bucket (consider a heavily clustered dataset). To ensure a fast query time, we want to bound the expected number of points in a bucket conditioned on $q$ landing in the bucket. This quantity is a function of the data and query distribution. Under the assumption that the data points are reasonably spread out (i.e., each hash bucket contains roughly the same number of database points), the bound of $n/2^k$ for a single table is approximately correct. To measure how "well-spread" the dataset $P$ is under the hash function $h : \mathbb{R}^d \to [2^k]$, we define the following quantity $M(P, h)$:

$$M(P, h) \stackrel{eq}{=} \sum_{i \in [2^k]} |h^{-1}(i) \cap P|^2 \ .$$

The table load $M(P, h)$ simply sums the squares of the hash bucket occupancy counts, which penalizes evenly spread out point sets less than highly clustered ones. With this definition, we can now analyze the expected number of spurious candidates under the assumption that the query $q$ comes from the same data distribution as the database $P$. In the regime where the mean correct angle between pre-softmax vectors and class vectors is small, this is a well justified assumption.

**Theorem 1.** *Suppose that the query point $q$ is chosen uniformly at random from the database $P$ and that the hash function $h$ was used to build the hash table. Then the expected number of points in the bucket containing $q$ is $M(P, h)/n$. Moreover, if the hash function values $\{h(p)\}_{p \in P}$ are independent random variables, then*

$$\mathbb{E}[M(P, h)] \ = \ M^* \ \stackrel{eq}{=} \ \frac{n^2}{2^k} + n \left(1 - \frac{1}{2^k}\right) \ . \tag{3}$$

*Proof.* For the first part, let $q$ be a point from $P$ chosen at random. The quantity of interest is

$$\mathbb{E}_q[\text{Number of points in bucket containing } q] = \sum_{q' \in P} \mathbb{E}_q[\mathbf{1}(q' \text{ and } q \text{ in same bucket})]$$

$$= \sum_{q' \in P} \mathbb{E}_q[\mathbf{1}(h(q) = h(q'))]$$

$$= \sum_{q' \in P} |h^{-1}(h(q')) \cap P|/n$$

$$= \sum_{i \in [2^k]} \sum_{q' \in P : h(q') = i} |h^{-1}(i) \cap P|/n$$

$$= \sum_{i \in [2^k]} |h^{-1}(i) \cap P|^2/n$$

$$= M(P, h)/n.$$

For the second part, we follow a similar chain of equalities and write:

$$M(P, h) = \sum_{i \in [2^k]} |h^{-1}(i) \cap P|^2$$

$$= \sum_{i \in [2^k]} \sum_{q' \in P : h(q') = i} |h^{-1}(i) \cap P|$$

$$= \sum_{q' \in P} |h^{-1}(h(q')) \cap P|$$

$$= \sum_{q' \in P} n \mathbb{E}_q[\mathbf{1}(h(q) = h(q'))].$$

Now note that $\mathbb{E}_q[\mathbf{1}(h(q) = h(q'))]$ is 1 when $q = q'$. Under the independence assumption, this expectation is $\frac{1}{2^k}$ in every other case. Thus we get

$$\mathbb{E}[M(P, h)] = \sum_{q' \in P} n \mathbb{E}_q[\mathbf{1}(h(q) = h(q'))]$$

$$= n + n(n-1)\frac{1}{2^k}$$

$$= \frac{n^2}{2^k} + n\left(1 - \frac{1}{2^k}\right).$$

The claim follows. $\qquad\qquad\qquad\qquad\qquad\qquad\qquad\qquad\qquad\qquad\qquad\qquad\qquad\qquad\qquad\quad\square$

If for a specific database $P$ and hash function $h$, $M(P, h)/M^*$ is close to 1, then the LSH table is sufficiently balanced for the inference to be efficient. Note that if we set $k$ to be $\lfloor \log_2 n \rfloor$, then $M/n$ is at most 3, i.e., we have a small constant number of candidate points per hash bucket. As a result, the number of candidates in the nearest neighbor search roughly equals the number of hash buckets. Using the earlier formula for the collision probability $p_\alpha$, we get the following formula for the number of candidate points as a function of angle $\alpha$:

$$\frac{5}{p_\alpha} = 5n^{-\log_2(1 - \frac{\alpha}{\pi})}$$

This expression allows us to estimate the number of candidate distance computations as we decrease the angle between the pre-softmax vector and the correct class vectors. We illustrate our estimate by substituting concrete numbers: if $n = 10^6$, improving $\alpha$ from $\pi/3$ (60 degrees) to $\pi/4$ (45 degrees) reduces the number of hash bucket lookups from about 16K to 1.5K, i.e., an order of magnitude improvement. In Figure 1(a), we plot the expression (C) as a function of the angle $\alpha$ for $n = 10^6$.

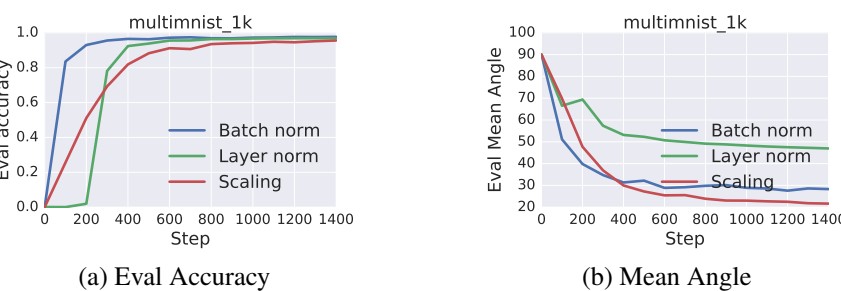

Figure 9: An example from multiMNIST_100k; label 73536.

| (a) Eval Accuracy | (b) Mean Angle |

Figure 10: Comparison of various Normalization Options on MultiMNIST 1K

# D    ADDITIONAL EXPERIMENTS

## D.1    EXPERIMENTS ON CELEBA

To investigate the impact of various network modifications on a third dataset, we also conduct experiments on CelebA (Liu et al., 2015). CelebA consists of about 200K face images of roughly 10K different celebrities. The goal of our experiment is to evaluate the quality of the representation learned by a standard Resnet model with 32 layers. In particular, we want to understand how well the learned representation *generalizes* to previously unseen identities (not only unseen images of celebrities that are also present in the training dataset). To this end, we learn a representation on about 8K identities via the multiclass approach that we also employed for other datasets. At evaluation time, we then take 2K unseen identities and compute representations for all examples (roughly 20K images). We compute two quantities: (i) the accuracy of a nearest-neighbor classifier on the evaluation dataset. (ii) the angular gaps to the nearest neighbor of the same class.

Our results show that normalizing the class vectors significantly improves both accuracy and angular gaps. Combining softmax normalization with the Batch Normalization – ReLU swap yields an accuracy of 74%, which is significantly higher than baselines without softmax normalization that achieve 60% accuracy (for both possible orders of normaliation layer an non-linearity). Moreover, the angle between query and nearest neighbor improves by about 8 degree while the set of evaluation points remains as balanced as the baselines without normalization of the class vectors.

## D.2    COMPARISON OF NORMALIZATION OPTIONS

We consider various options for controlling the norms of the representation vectors.

BN: Batch Normalization.

LN: Layer Normalization (Ba et al., 2016).

Scaling: Using a normalized representation $\hat{\phi}_\theta(\mathbf{x}) = \phi_\theta(\mathbf{x})/\|\phi_\theta(\mathbf{x})\|$.

To evaluate these options, we used multiMNIST data set with 1K classes. We use a network identical to that in Section 4 with a representation layer of 256 dimensions. We compare these options on various axes: the final accuracy, the speed of convergence and the mean angle. For each of the configurations, we did a grid search over learning rates to pick the best one. The final accuracy and angles are shown in Figure 10. As the figures show, the network with batch norm gives nearly the same angles as the best of these options. In terms of convergence speed and accuracy, batch norm is the noticeably better than the other options.

