# OpenReview forum: "Learning Representations for Faster Similarity Search"
_ICLR.cc/2018/Conference — Reject_

### Official Review · AnonReviewer2 · 2017-11-24
**Initial review**

**Rating:** 4
**Confidence:** 5

**Review:**


The context is indexing images with descriptor vectors obtained from a DNN. This paper studies the impact of changing the classification part on top of the DNN on the ability to index the descriptors with a LSH or a kd-tree algorithm. The modifications include: applying or not a ReLU and batch normalization (and in which order) and normalizing the rows of the last FC layer before softmax.

+ : preparing good features for indexing has not been studied AFAIK, and empirical studies that pay attention to details deserve to be published, see eg. the series "all about vlad" and "Three things everyone should know to improve object retrieval" by Arandjelović, Zisserman

+/- : the paper considers two indexing methods (kd-tree and LSH) but basically it evaluates how well the features cluster together in descriptor space according to their class. Therefore it should be applicable to more SOTA techniques like product quantization and variants.

- : there is no end-to-end evaluation, except figure 7 that is not referenced in the text, that that has a weird evaluation protocol ("the probability of finding the correct nearest neighbor conditioned on the model being correct")

- : the evaluation context is supervised hashing, but the evaluation is flawed: when the same classes are used for evaluation as for training, there is a trivial encoding that consists in encoding the classifier output (see "How should we evaluate supervised hashing?" Sablayrolles et al).

- : no comparison with the SOTA, both for the experimental setup and actual results. There are plenty of works that do feature extraction + indexing, see "Deep Image Retrieval: Learning global representations for image search", Gordo et al ECCV'16, "Neural codes for image retrieval", Babenko et al ECCV'14, "Large-Scale Image Retrieval with Attentive Deep Local Features", Noh et al ICCV'17, etc.


Details:

example x comes from a domain X typically R^p --> X = R^p but is it an image or an embedding? does the training backprop on the embedding?

top equation of p4 -> if you don't use it don't introduce it

section 3.2 is a bit disappointing. After 3.1, the natural / "ML-correct" way of handing this would be to design a loss that enforces the desirable properties, but this is just a set of tricks, albeit carefully justified

Section 4 could use some clarification. Is the backprop applied to the embedding ("we convert each frame to its VGG features")? "d taken between 256 and 1024": which one?

More importantly, the whole section looks like a parametric evaluation with an intermediate objective (mean angle).


Overall I think the paper does not have a significant enough contribution or impressive enough results to be published.

---

### Official Review · AnonReviewer1 · 2017-11-27
**Lacking focus**

**Rating:** 4
**Confidence:** 5

**Review:**

This paper investigates learning representations for the problem of nearest neighbor (NN) search by exploring various deep learning architectural choices. The crux of the paper is the connection between NN and the angles between the closest neighbors -- the higher this angle, more data points need to be explored for finding the nearest one, and thus more computational expense. Thus, the paper proposes to learn a network that tries to reduce the angles between the inputs and the corresponding class vectors in a supervised framework using softmax cross-entropy loss. Three architectural choices are investigated, (i) controlling the norm of output layers of the CNN (using batch norm essentially), (ii) removing relu so that the outputs are well-distributed in both positive and negative orthants, and (iii) normalizing the class vectors. Experiments are given on multiMNIST and Sports 1M and show improvements.

Pros:
1) The paper explores different architectural choices for the deep network to some depth and show extensive results.
2) The results do demonstrate clearly the advantage of the various choices and is useful
3) The theoretical connections between data angles and query times are quite interesting,

Cons:
1) Unclear Problem Statement.
I find the problem statement a bit vague. Standard NN search finds a data point in the database closest to a query under some distance metric. While, the current paper uses the cosine similarity as the distance, the deep framework is trained on class vectors using cross-entropy loss. I do not think class labels are usually assumed to be given in the standard definition of NN, and it is not clear to me how the proposed setup can accommodate NN without class labels.  Thus as such, I see this paper is perhaps proposing a classification problem and not an NN problem per se.

2) Lacks Focus
The paper lacks a good organization in my opinion. Things that are perhaps technically important are moved to the Appendix. For example, I find the theoretical part of the paper (e.g., Theorem 1) quite elegant and perhaps the main innovation in this paper. However, that is moved completely to the Appendix. So it cannot be really considered a contribution. It is also not clear if those theoretical results are novel.

2) Disconnect/Unclear Assumptions
There seems to be some disconnect between LSH and deep learning architectures explored in Sections 2 and 3 respectively. Are the assumptions used in the theoretical results for LSH also assumed in the deep networks? For example, as far as I know, the standard LSH works assumes the projection hyperplanes are randomly chosen and the theoretical results are based on such assumptions. It is not clear how a softmax output of a CNN, which is trained in a supervised way, follow such assumptions. It would be important if the paper could clarify such assumptions to make sure the sections are congruent.

3) No Related Work
There have been several efforts for adapting deep frameworks into KNN. The paper ignores all such works. Thus, it is not clear how significant is the proposed contribution. There are also not comparisons what-so-ever to competitive prior works.

4) Novelty
The main contribution of this paper is basically a set of experiments looking into architectural choices. However, the results of this study do not provide any surprises. It appears that batch normalization is essential for good performances, while using RELU is not so when one wants to use all directions for effective data encoding. Thus, as such, the novelty or the contributions of this paper are minor.

Overall, while I find there are some interesting theoretical bits in this paper, it lacks focus, the experiments do not offer any surprises, and there are no comparisons with prior literature. Thus, I do not think this paper is ready to be accepted in its present form.

---

### Official Review · AnonReviewer3 · 2017-11-27
**Interesting work; need some rewriting to clarify the problem(s) being solved and how**

**Rating:** 4
**Confidence:** 4

**Review:**

The authors are trying to improve the efficiency of similarity search on representations learned by a deep networks but it is somewhat unclear where the proposed solution will be applied and how. The idea of modifying the network learning process to obtain representations that allow for faster similarity search has definitely a lot of value. I believe that the manuscript needs some re-writing so that the problem(s) are better motivated and is easier to follow.

Specific comments:
- Page 2, Sec 2.1, 2.2: The theoretical/empirical analysis is Section 2 has actually been properly formalized by Sanjiv Kumar, et al. [a], and Kaushik Sinha, et al.[b]'s papers on relative contrast and related quantities [a]. It would be good to discuss the proposed quantity in reference to these existing quantities. The idea presented here appears too simplistic relative to the existing ones.
- Page 2, Sec 2.1: Usually in NNS, the data is not "well-spread" and has an underlying intrinsic structure. And being able to capture this intrinsic structure is what makes NNS more efficient. So how valid is this "well-spread"-ness assumption in the setting that is being considered? Is this common in the "learned representations" set up?
- Page 4, After Eq 2: I think the properties 1,2 are only true if you are using softmax to predict label and care about predictive 0-1 accuracy. Is that the only place the proposed solution is applicable or am I misunderstanding something?
- Figure 2, 4 and 7 don't seem to be referred anywhere.
- Is the application of NNS in performing the softmax evaluation? This needs to be made clearer.
- If the main advantage of the proposed solution is the improvement of training/testing time by solving angular NNS (instead of MIPS) during the softmax phase, a baseline using existing MIPS solution [c] need to be considered to properly evaluate the utility of the proposed solution.

[a] He, Junfeng, Sanjiv Kumar, and Shih-fu Chang. "On the Difficulty of Nearest Neighbor Search." Proceedings of the 29th International Conference on Machine Learning (ICML-12). 2012.

[b] Dasgupta, Sanjoy, and Kaushik Sinha. "Randomized partition trees for exact nearest neighbor search." Conference on Learning Theory. 2013.

[c] Neyshabur, Behnam, and Nathan Srebro. "On Symmetric and Asymmetric LSHs for Inner Product Search." Proceedings of the 32nd International Conference on Machine Learning (ICML-15). 2015.

---

### Decision · Program_Chairs · 2018-01-29
**ICLR 2018 Conference Acceptance Decision**

**Decision:**

Reject

**Comment:**

The paper received three good quality reviews which were in agreement that the paper was below the acceptance threshold. The authors are encouraged to follow the suggestions from the reviews to revise the paper and resubmit to another venue.